# Hot Oscillatory Pressing of Carbon Nanotube-Reinforced Copper Matrix Nanocomposite

**DOI:** 10.3390/nano11092411

**Published:** 2021-09-16

**Authors:** Min Han, Yunpeng Ding, Jinbiao Hu, Zhiai Shi, Sijia Jiao, Xiaoqin Guo, Hanying Wang, Linan An

**Affiliations:** 1School of Materials, Zhengzhou University of Aeronautics, Zhengzhou 450046, China; hanmin188@126.com (M.H.); jinbiao2021@126.com (J.H.); szhiai@126.com (Z.S.); jiaosijia@126.com (S.J.); hanyingwang@126.com (H.W.); 2Department of Materials Science and Engineering, University of Central Florida, Orlando, FL 32816, USA; linan74@gmail.com

**Keywords:** carbon nanotube, copper matrix nanocomposite, hot oscillatory pressing, densification, hardness

## Abstract

Carbon nanotube reinforced copper matrix nanocomposites have great potential in machinery, microelectronics, and other applications. The materials are usually prepared by powder metallurgy processes, in which consolidation is a key step for high performance. To improve the density and mechanical properties, the authors explored the use of hot oscillatory pressing (HOP) to prepare this material. A carbon nanotube reinforced copper matrix nanocomposite was synthesized by both HOP and hot pressing (HP) at various temperatures, respectively. The samples prepared by HOP exhibited significantly higher density and hardness than those prepared by HP at the same temperature, and this was because the oscillatory pressure of HOP produced remarkable plastic deformation in copper matrix during sintering. With the decrease of sintering temperature in HOP, the amount of deformation defect increased gradually, playing a key role in the increasing hardness. This work proves experimentally for the first time that HOP can produce much more plastic deformation than HP to promote densification, and that HOP could be a very promising technique for preparing high-performance carbon nanotube reinforced copper matrix nanocomposites.

## 1. Introduction

Carbon nanotube reinforced copper matrix nanocomposites (CNT/Cu nanocomposites) have great potential in machinery, microelectronics, new energy, and other applications due to the high strength and conductivity as well as good chemical stability [1,2]. The materials are usually prepared by powder metallurgy processes, in which consolidation is a key step that has a great effect on the microstructures and properties of resultant materials [3,4]. Due to the poor wettability between CNT and Cu [5], consolidation of CNT/Cu nanocomposites is not as easy as it seems, especially when the CNT content is high. When conventional sintering techniques are used, long sintering time and high sintering temperature are required for densifying the materials [6]. The resultant materials usually exhibited low relative density and coarse-grained structures; thus, they required subsequent deformation (e.g., forging, extrusion, and rolling [7]) to further increase density [8,9]. However, the subsequent deformation can damage the structure of CNTs and the properties of the material [10].

In recent years, to improve the density and mechanical property, CNT/Cu nanocomposites were also prepared using some new sintering technologies, such as spark plasma sintering (SPS) [11], microwave sintering (MS) [12], and laser sintering (LS) [13,14]. Zhang et al. [11] showed that SPS can significantly shorten the sintering time, likely by improving the surface activity of copper powder and enhancing atomic diffusion. They prepared the nanocomposite containing 3 vol.% of CNTs with density up to 98.9% and inhibited grain growth, which exhibited a strength nearly double that of the matrix. Duan et al. [12] prepared CNT/Cu composites with density of 95% and Vickers hardness of 80 by microwave sintering at 1000 °C. Stasic et al. [13] and Gu et al. [14] used high energy laser sintering to prepare high-strength particle-reinforced copper matrix composites with density up to 98% and 90.7%, respectively. Except for SPS, other new techniques did not lead to significant improvement in preparation of CNT/Cu nanocomposites.

Recently, a unique sintering technology—hot oscillatory pressing (HOP, or oscillatory pressure sintering)—was developed [15], which uses oscillatory pressure to replace constant pressure in traditional hot pressing (HP). HOP was applied to many ceramic materials such as WC [16], zirconia [17], alumina [18], silicon nitride [19], aluminum nitride [20], and high entropy ceramic [21], as well as refractory alloy [22] and cemented carbide [23,24]. HOP can enhance densification, inhibit grain growth, and improve mechanical properties of these materials. Several possible mechanisms were speculated to explain such improvements, including that the oscillatory pressure of HOP can promote particle rearrangement, enhance the sliding of grain boundary, and generate plastic deformation [25,26,27], though there is little experimental evidence about these mechanisms.

In this paper, we report for the first time the preparation of CNT/Cu nanocomposites by HOP. Compared with that of HP, HOP can significantly improve the density and mechanical property of the resultant materials. We demonstrate that such improvements are credited to the oscillatory pressure of HOP, which can generate remarkable plastic deformation during sintering.

## 2. Materials and Methods

Commercially available copper powder (purity 99.9%, diameter 0.5 μm, Nanou Co., Ltd., Shanghai, China) and CNT (outer diameter 30–50 nm, length 10–20 μm, Chinese Academy of Sciences, Chengdu Organic Chemistry Co., Ltd., Chengdu, China) were employed as the original materials. The received CNTs were coated by Cu using electroless plating method [28]. The received Cu powder and 3 vol.% of Cu-coated CNTs were placed in a steel vial and then mixed by ball milling under argon atmosphere in the mill at 300 rpm for 2 h. Zirconia balls of 5 mm were employed as the milling medium and the ball-to-material weight ratio was 10:1. The mixture of powder was then placed into a cylinder-shaped graphite die with an inner radius of 7.5 mm and sintered in the hot oscillatory pressing system (OPS 2020, Efield Materials Technology Co., Ltd., Chengdu, China). The sintering schedule of HOP was as follow. Firstly, the samples were heated to the sintering temperature (550, 600, 650 and 700 °C) at a heat-up rate of 8 °C/min with a pressure of 40 MPa in a vacuum (<10 Pa). Then, the sample was held at the sintering temperature for 60 min under either an oscillatory pressure of 40 ± 10 MPa with a frequency of 5 Hz (HOP) or an invariable pressure of 40 MPa (HP). Finally, the samples were cooled down to RT with a cooling rate of 15 K/min. The detailed sintering schedule of HOP and HP is shown in Figure 1.

Relative density was the ratio of measured density to theoretical density, which was used to characterize the degree of densification in materials. The Archimedes method was adopted to measure the density of sintered samples. The theoretical density was the sum of the product of the density and corresponding volume fraction for each component in the composite. The microhardness tests were carried out on the polished samples under a load of 500 g for 15 s. Then, the samples were characterized by scanning electron microscope (SEM, JSM-7001F, JEOL, Tokyo, Japan), electron back-scattered diffraction (EBSD, HKL Nordlys Nano, Oxford, Oxford, UK) and transmission electron microscope (TEM, JEM-2100, JEOL, Tokyo, Japan). The SEM samples were prepared by etching the polished surface of the sintered samples. Before EBSD test, the surface of samples was mechanically polished and electropolished in perchloric acid solution. During EBSD test, for each specimen, an area of 20 × 20 μm were scanned with a step size of 60 nm. Channel 5 software (5.1, Oxford, Oxford, UK) was employed as the analysis software. During EBSD analysis, the average value of MAD (mean angular deviation) of each specimen was as follows: 0.47 (HOP 550 °C), 0.43 (HOP 600 °C), 0.45 (HOP 650 °C), 0.43 (HOP 700 °C), 0.4 (HP 550 °C), and 0.32 (HP 700 °C), respectively. TEM specimens were prepared by grinding and ion polishing.

## 3. Results

### 3.1. Microstructure, Density and Hardness

Figure 2 exhibits the morphology of the CNTs, Cu-coated CNTs, Cu powder, and mixture of Cu and CNT powders. As shown in Figure 2a, the outer diameter of as-received CNTs is about 30–50 nm. After the process of electroless plating, copper particles are attached to the surface of as-received CNTs (Figure 2b), which is beneficial to the future processing. The as-received Cu powder have a spherical shape, with a diameter of 0.2–1 μm, and the mean diameter is about 0.5 μm (Figure 2c). After ball milling, the CNTs are evenly embedded into the copper particles (Figure 2d) by the impact and friction processes during milling [29], which can help to make the distribution of CNTs and Cu powder more uniform. The length of CNTs is about 1 μm in the mixture, much shorter than their as-received length, illustrating that ball-milling significantly cut the CNTs.

Figure 3 shows the relative density of samples sintered by HOP and HP at different sintering temperatures. Compared with that of HP, HOP can significantly improve the densification of the CNT/Cu nanocomposite. While the density of the samples prepared by both techniques increases as the sintering temperature increases, the sample prepared by HOP exhibits much higher density than that of the sample prepared by HP at the same sintering temperature. The sample prepared by HOP at 550 °C has the same density as the sample prepared by HP at 650 °C, which indicates that the oscillatory pressure can remarkably reduce the sintering temperature. The density of the sample prepared by HOP at 700 °C reaches 99.5%, which is higher than most similar materials prepared by other techniques reported in the literature [12,30,31,32].

The microstructure of the obtained samples was characterized by SEM, EBSD, and TEM. Figure 4 shows the typical results obtained from the sample prepared by HOP at 550 °C. 

Figure 4a,b reveal that the composite exhibits a dense and uniform microstructure without discontinuous grain growth. The average grain size of the matrix measured from SEM images is ~0.60 μm, which is the same as the particle size of the original copper powder, suggesting that there is almost no grain growth during densification. Figure 4c is the SEM image obtained from the over-etched polished surface, which reveals that CNTs are uniformly distributed in the matrix.

Figure 4d is the TEM of a CNT within the matrix, and the CNT exhibits good integrity. Besides length reduction, CNTs were not further damaged by ball milling and consolidation. The interface between the CNT and the matrix is clean, indicating no reaction between the two. Comparing the results obtained from different samples shows that the microstructure of all samples is almost identical regardless of processing conditions.

The mechanical behavior of the obtained samples was investigated by measuring their hardness. The result (Figure 5) shows that the sample prepared by HOP exhibits much higher hardness than that of the sample prepared by HP at the same temperature, suggesting HOP can improve the mechanical behavior of the nanocomposite. The highest hardness is observed for the sample prepared by HOP at 550 °C, reaching 159 HV. This value is much higher than that observed for the samples prepared by HP at 650 and 700 °C even though they have the same density and microstructure. The value is also higher than that of the previously reported CNT/Cu nanocomposites with the similar CNT concentration [12,30,31,32].

Moreover, the authors compare the relative density and hardness of Cu-based materials fabricated by different sintering methods in literatures [12,30,31,33,34] and the current nanocomposite sintered by HOP exhibits higher relative density and hardness than any others. In particular, the HOP sample shows higher hardness than that of the SPS sample [33,34], though with similar relative density. Furthermore, in Figure 5, the hardness of the sample prepared by HP increases with increasing sintering temperature, which is easy to understand since the samples exhibit the same microstructure but higher density as sintering temperature increases. However, there is no significant difference in the hardness of the samples sintered by HOP at different temperatures, even though the density of sample increases with temperature.

### 3.2. Distribution of Grain Boundaries and Deformation Defect

To uncover the mechanisms responsible for the observed densification and mechanical behavior, the samples were further analyzed using EBSD in detail (Figure 6a–f). The blue, orange, black, and red lines represent the grain boundaries with the misorientation angles less than 2° (subgrain boundaries, SGBs), the misorientation angles between 2 to 10° (low-angle grain boundaries, LAGBs), the misorientation angles > 10° (high-angle grain boundaries, HAGBs), and the misorientation angle of 60° (twin boundaries, TBs), respectively. The sample prepared by HOP contains much more small-angle grain boundaries (SAGBs, including SGBs and LAGBs) than that of the sample prepared by HP. 

Figure 6g plots the distribution of grain boundaries as a function of misorientation angles for the sintered samples. The plots confirm that the samples prepared by HOP contain much more SAGBs than that of the sample prepared by HP. Figure 6h illustrates the effect of sintering temperature on the concentration of SAGBs. The sample prepared by HOP contains more than 5 times the amount of SAGBs than the samples prepared by HP. The SAGBs content of the samples prepared by HOP and HP decreases as sintering temperature increases. Figure 6i is a plot of TBs content as a function of sintering temperature, which reveals that the concentration of TBs in the sample prepared by HOP and HP is almost the same and remains unchanged with the increase of sintering temperature.

Figure 7 are deformation fraction maps of the samples prepared by HOP and HP at different temperatures, respectively. The area of plastic deformation in the sample prepared by HOP is much larger than that of the sample prepared by HP. The fraction of deformation area was measured from EBSD maps and plotted as a function of sintering temperature in Figure 8. The deformation area of the samples prepared by HOP is 4 to 8 times that of the samples prepared by HP. The deformation area in the samples prepared by HOP and HP decreases as sintering temperature increases.

The plastic deformation in the nanocomposite was further studied by TEM, as shown in Figure 9. The results reveal that the samples prepared by HOP contain a larger number of lattice defects, such as dislocations (Figure 9a) and dislocation walls (Figure 9b), while such dislocation activity was not observed from the samples prepared by HP (Figure 9c). This indicates that the oscillatory pressure of HOP produces much more dislocations than the static pressure of HP.

## 4. Discussion

The above results (Figure 6, Figure 7, Figure 8 and Figure 9) clearly reveal that compared with that of HP, HOP produced much more plastic deformation via the generation and motion of dislocations, instead of twinning. This can be rationalized as follows. The dislocation density should be proportional to the plastic deformation strain that happened during densification. Although the apparent plastic strain, which is proportional to the increases in density during sintering (assume the fraction of deformation strain is the same for all samples), is similar for the samples prepared by HOP as for the samples prepared by HP, the “real” total plastic strain in the samples prepared by HOP should be much greater than in those prepared by HP due to the repeated deformations caused by the oscillatory pressure of HOP. The decrease in plastic deformation with increasing sintering temperature (Figure 7) is likely due to the balance between dislocation formation and recrystallization. The high dislocation density produced by deformation can provide driving force and nucleation sites for recrystallization at high temperature, which leads to the removal of lattice defects. In the temperature rang tested here, the balance shifts to the recrystallization as the sintering temperature increases, resulting in the decrease in the density of lattice defects (Figure 6 and Figure 7).

The above results can also be used to rationalize the trend of hardness changes observed from different samples. Since the samples prepared by HOP exhibit significant plastic deformation, it is reasonable to expect that the hardness of the samples is determined by the degree of plastic deformation and their relative density. On the other hand, the hardness of the samples prepared by HP is determined by their relative density. Thus, the increase in hardness exhibited by the samples prepared by HOP (Figure 5) is due to the strain hardening effect. For example, although they have the same density and grain size, the samples prepared by HOP at 550 °C have higher hardness than those prepared by HP at 650 °C since the former has greater plastic deformation. The hardness of the samples prepared by HP increases with higher temperature (Figure 5). This is due to the increase of relative density with higher temperature (Figure 3). On the other hand, the weak effect of sintering temperature on the hardness of the samples prepared by HOP (Figure 5) is the trade-off between the degree of plastic deformation, which decreases with temperature and relative density, which increases with temperature (Figure 3).

From the above, this paper proves for the first time that the improvements in densification and mechanical properties exhibited by HOP-prepared materials are due to the plastic deformation produced by the oscillatory pressure, as speculated by the literatures [25,26,35].

## 5. Conclusions

In this paper, CNTs/Cu nanocomposite with 3 vol.% of multiwall carbon nanotubes was prepared by both hot oscillatory pressing (HOP) and hot pressing (HP) at different temperatures ranging from 550 to 700 °C. The results led to the following conclusions:(1)Compared to that of HP, the samples prepared by HOP exhibited significantly higher density and hardness.(2)The improved densification of HOP is because the oscillatory pressure caused plastic deformation, and the improved hardness is due to the strain-hardening effect.(3)With the decrease of sintering temperature in HOP, the amount of deformation defects increased gradually, playing a key role in the increasing hardness.

This work proves experimentally for the first time that HOP can produce much more plastic deformation than HP to promote densification and improve hardness. Therefore, the current study suggests that HOP is a promising technique for the preparation of high-performance CNT/Cu nanocomposites.

## Figures and Tables

**Figure 1 nanomaterials-11-02411-f001:**
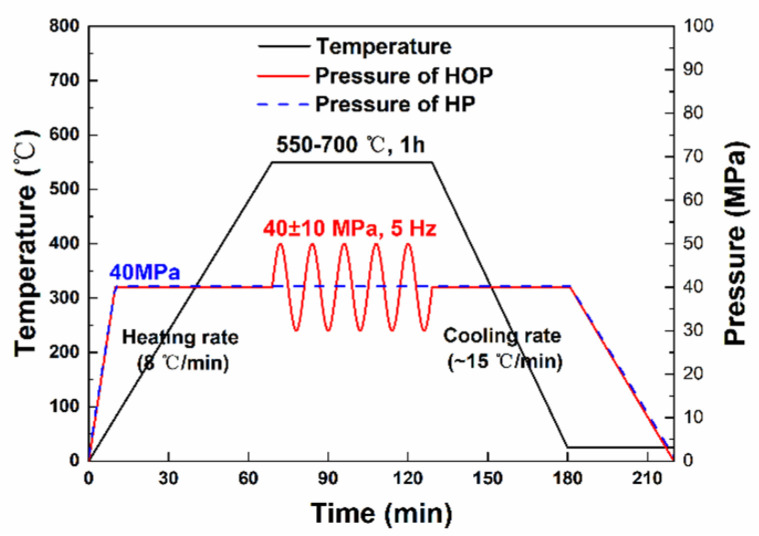
Detailed sintering schedule of HOP and HP.

**Figure 2 nanomaterials-11-02411-f002:**
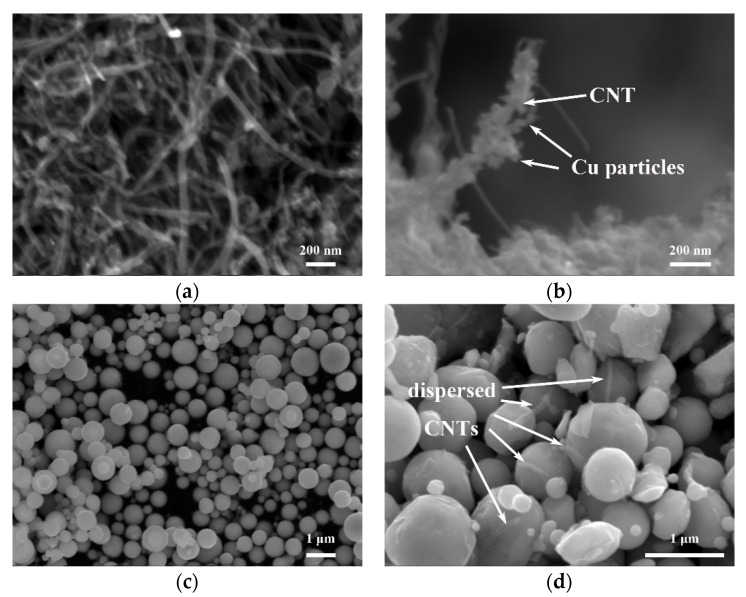
SEM images of (**a**) as-received CNTs, (**b**) Cu-coated CNTs, (**c**) pure Cu powder, and (**d**) mixture of CNT and Cu powders.

**Figure 3 nanomaterials-11-02411-f003:**
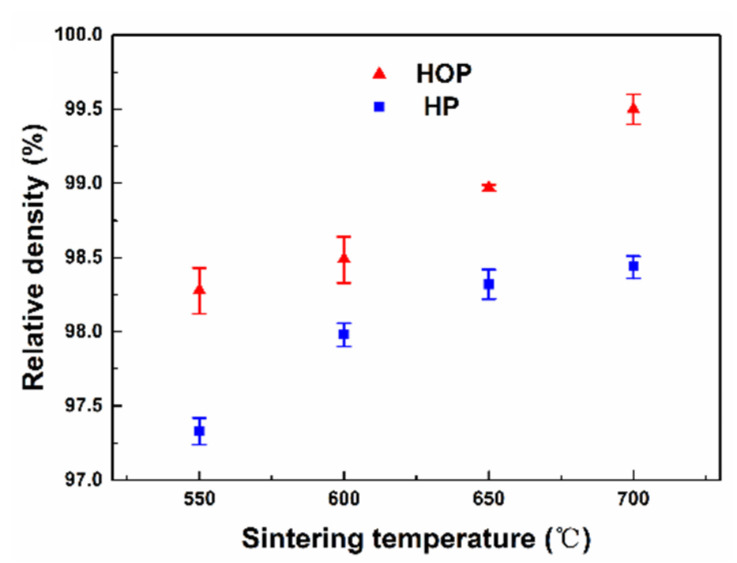
A plot of relative density as a function of sintering temperature for samples sintered by HOP and HP, respectively.

**Figure 4 nanomaterials-11-02411-f004:**
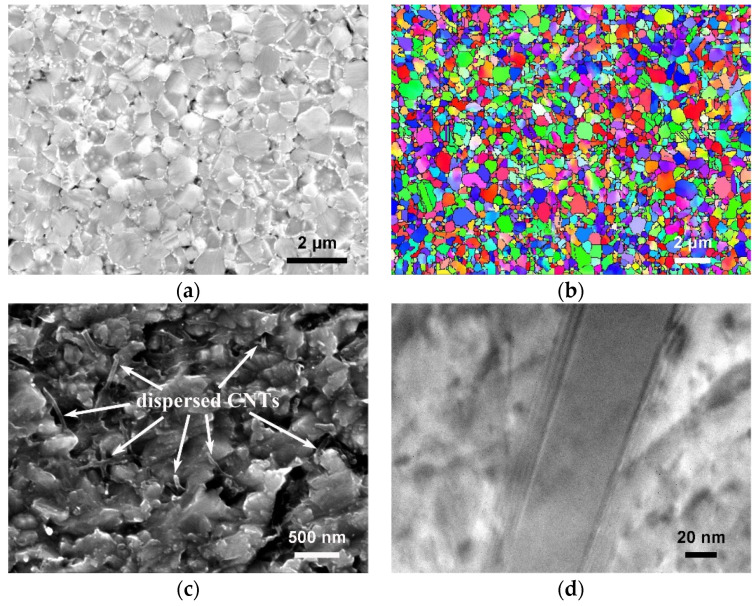
Microstructure of sample prepared by HOP at 550 °C. (**a**) SEM and (**b**) EBSD images showing microstructure. (**c**) SEM image of over-etched sample showing distribution of CNTs. (**d**) TEM image of a CNT in the Cu matrix.

**Figure 5 nanomaterials-11-02411-f005:**
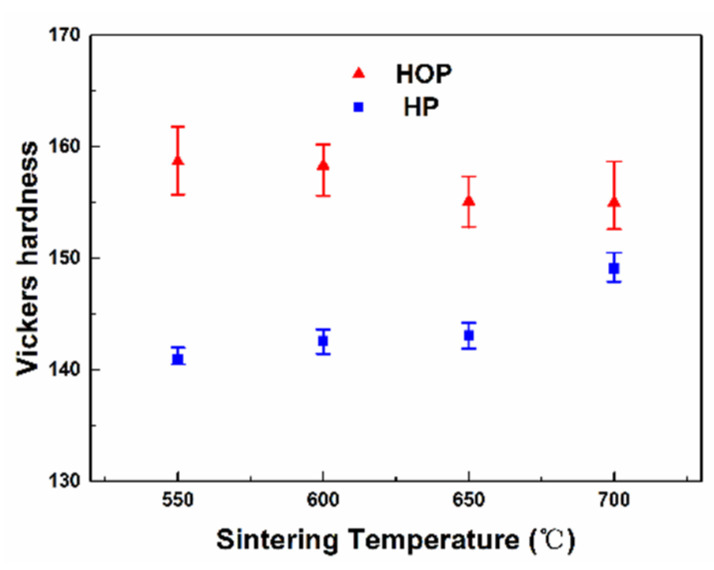
A plot of hardness as a function of sintering temperature for samples sintered by HOP and HP.

**Figure 6 nanomaterials-11-02411-f006:**
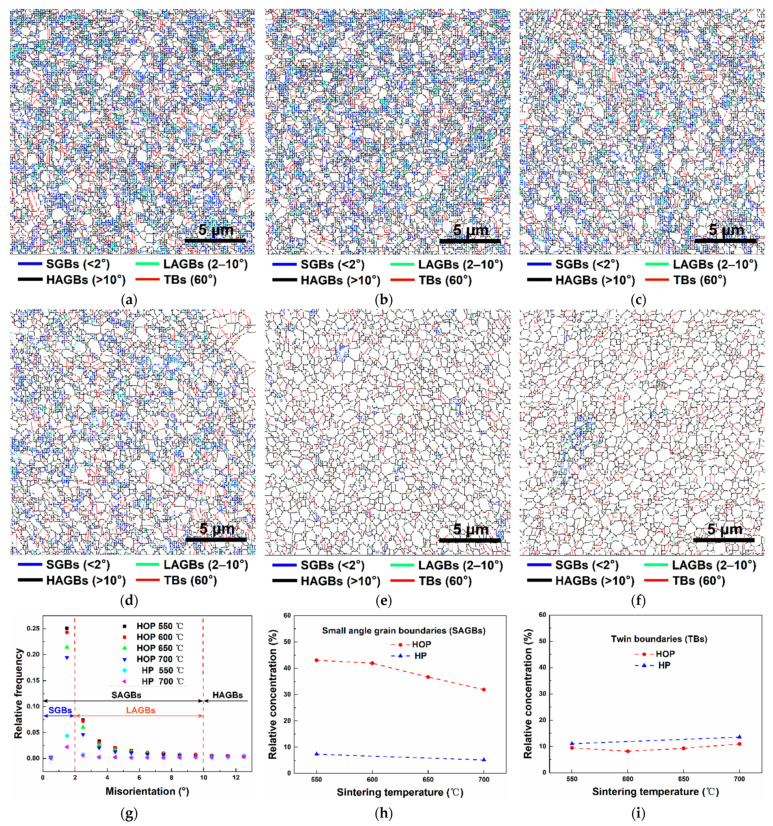
(**a**–**f**) EBSD maps of grain boundary for samples prepared by (**a**) HOP at 550 °C, (**b**) HOP at 600 °C, (**c**) HOP at 650 °C, (**d**) HOP at 700 °C, (**e**) HP at 550 °C, and **(f**) HP at 700 °C, respectively. (**g**) Probability distribution of grain boundaries as a function of misorientation angles for sintered samples. (**h**,**i**) Relative concentration of small–angle grain boundaries and twin boundaries as a function of sintering temperature, respectively, for samples sintered by HOP and HP, respectively.

**Figure 7 nanomaterials-11-02411-f007:**
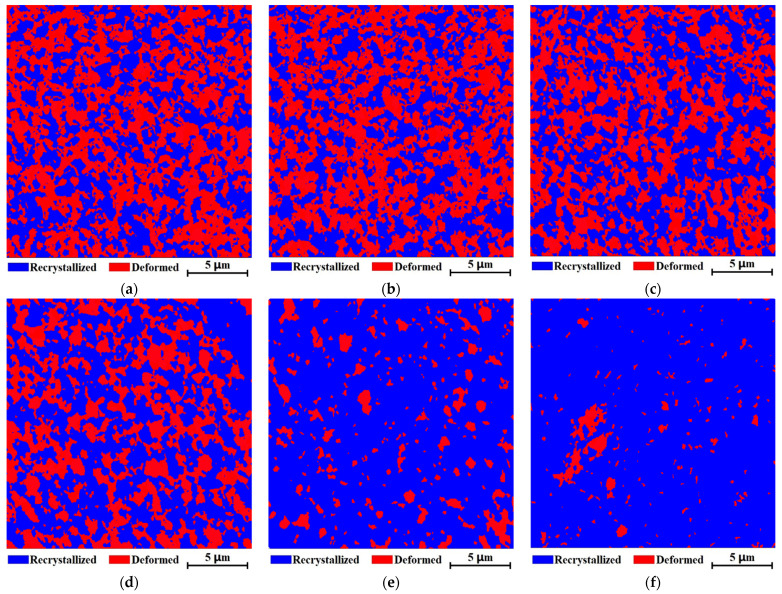
Deformation fraction maps of samples prepared by (**a**) HOP at 550 °C, (**b**) HOP at 600 °C, (**c**) HOP at 650 °C, (**d**) HOP at 700 °C, (**e**) HP at 550 °C, and (**f**) HP at 700 °C, respectively.

**Figure 8 nanomaterials-11-02411-f008:**
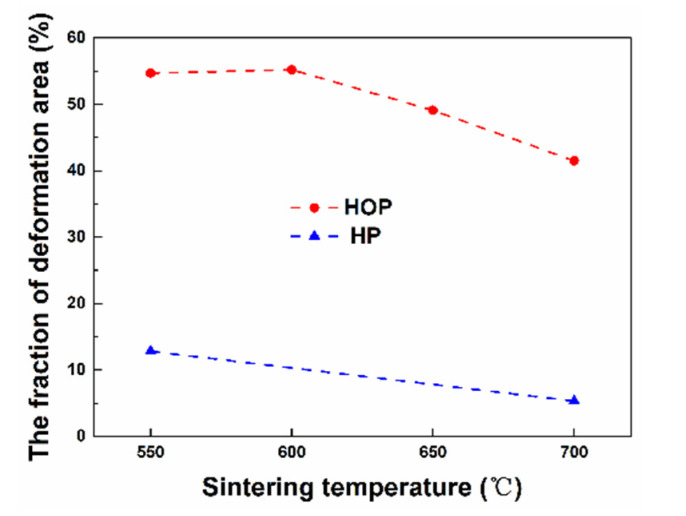
Fraction of deformation area as a function of sintering temperature for samples sintered by HOP and HP.

**Figure 9 nanomaterials-11-02411-f009:**
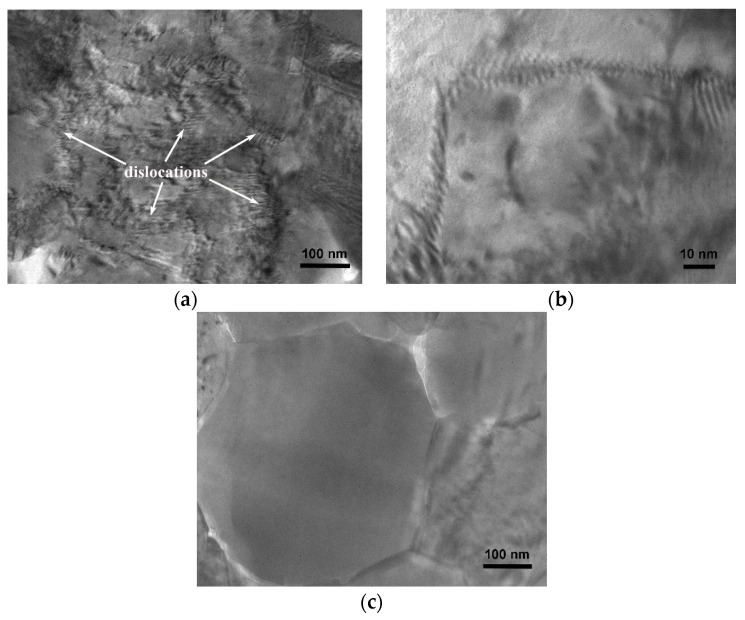
TEM images of samples prepared by (**a**,**b**) HOP and (**c**) HP at 550 °C.

## Data Availability

The data presented in this study are available in Appendix A.

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
