# Peer review of "Hot Oscillatory Pressing of Carbon Nanotube-Reinforced Copper Matrix Nanocomposite"

_nanomaterials, 2021, doi:10.3390/nano11092411_

Round 1

Reviewer 1 Report

The topic of this paper, development of new technique for making high performance nanocomposites, is suitable for publication in this journal. However, there are mandatory revisions that must be made before the paper can be accepted for publication. Those are as follows:

Materials and Methods, Line 91

The authors should describe the detailed conditions of EBSD analysis.

Figure 3

How did the authors calculate the theoretical density?

Figures of 2, 4, 7, 8 and 10

The authors must correct the positions of symbols (such as (a) or (b)). It is difficult to understand which one it shows.

Figure 7

It is difficult to distinguish between the red and orange lines, so change the color of either line.

Reviewer 2 Report

The paper presents the effect of using Hot Oscillatory Pressing (HOP) to prepare copper/carbon nanotubes composites, as far as I am aware the work is novel and the results provide important information as to why HOP helps to densify and increase the hardness of metals compared to other sintering technologies . The manuscript is well structured and clearly written. However, there are some aspects that need modification before the article can be published.

  1. In the introduction (line 50), it would be useful if other names of the HOP are given. For example "oscillatory pressure sintering".
  2. In the introduction (Line 60), the word "time" is missing, so the phrase would read "we report for the first time the preparation..."
  3. In the methodology (Line 81), please clarify what does it mean "cool down to room temperature with furnace". Was the sample cooled down inside the furnace by natural convection or the furnace has a cooling mechanism to achieve 15 K/min?
  4. Please add error bars to the data shown in Figures 3, 6, 7 and 9. Also explain what kind of mathematical function are the trend line shown in Figure 3, 7 and 9.
  5. In line 125, there is a strange symbol (a spiral), probably instead of micrometer. Please correct it.
  6. In line 133, it is mentioned that ball milling did not damaged the CNT, but it was mentioned  before that they were cut. Therefore, it is suggested to mention that besides length reduction CNTs were not further damaged by ball milling and consolidation.
  7. Please provide evidence that : Comparing the results obtained from different samples shows that the microstructure of all samples is almost identical regardless of processing conditions. (line 134 and 135).
  8. In Figure 5, please mention what type of function is the trend line
  9. In line 156, it is mention that "the hardness of the sample prepared by HOP is slightly decreases as sintering temperature increases".
    However, all values are within the error bars of the measurement points, therefore the difference is not significant and it cannot be claimed that the hardness decreases with sintering temperature for HOP. Please correct this statement.
  10.  In section 3.2, please correct the degree symbol in the text. It should be ° not o.
  11. In Figure 10, please indicate on the images where the dislocations are, that way it will be obvious and clearer.
  12. In line 224, it is mentioned that "the hardness of the samples is determined by the degree of plastic deformation and their relative density"; therefore it would be useful to show the relationship between the degree of plastic deformation, hardness and density (maybe as a 3D plot or a radar/spider chart) since you have the data available anyway.
  13. In the conclusion (line 251) please include that HOP also improves hardness due to plastic deformations.

Round 2

Reviewer 1 Report

The authors should also describe MAD value for EBSD analysis.
